



# Investigating seasonal and multi-decadal water/ice storage changes in the Murtèl rock glacier using time-lapse gravimetry

Landon J.S. Halloran[1] and Dominik Amschwand[2]

[1]Centre for Hydrogeology and Geothermics (CHYN), University of Neuchâtel, Switzerland
[2]Networked Embedded Sensing Center, Department of Computer Science, University of Innsbruck, Austria

**Correspondence:** Landon Halloran (landon.halloran@unine.ch)

**Abstract.** Rock glaciers are important features of many alpine hydrological systems. Although their seasonal release of water enhances the resilience of alpine headwater catchments to climate change, measurement of their internal water and ice storage changes remains a challenge. Recent technological and methodological advances have enabled novel applications of time-lapse gravimetry (TLG) to estimate subsurface storage changes. Here, we present the first application of TLG on a rock glacier. We measure seasonal (July–Sept) changes in gravity at the Murtèl rock glacier (Upper Engadine, Switzerland). We employ drone-based photogrammetry to correct for surface mass changes in the form of snow. We also compare the *Bouguer* anomaly of our 2024 surveys with those from a pioneering 1991 gravimetry study. The seasonal results reveal spatial differences in active layer thaw, with estimated ice storage loss ranging between $11-64$ cm water equivalent, while the multi-decadal results suggest zonal decreases in permafrost ice storage. Our study provides new insights into rock glacier–groundwater processes and illustrates how TLG can be employed to measure cryospheric and hydrogeological processes in permafrost and periglacial landforms.

## 1  Introduction

Our changing climate is imparting profound shifts to the hydrological regime of high-mountain regions (Hock et al., 2022). Temperature, evaporative losses, and precipitation variability are increasing, which could lead to more frequent droughts and reduced streamflow in dry years and thus threaten downstream water security (Schaffer et al., 2019; Hoelzle et al., 2020; Barandun et al., 2020; Arenson et al., 2022). As glaciers recede (Hugonnet et al., 2021) and precipitation in the form of snow decreases (Gottlieb and Mankin, 2024), the dynamics of many high mountain regions are shifting from glacial to periglacial regimes (Haeberli et al., 2017). In this context of reduced surface snow and ice volumes and increasing risk of precipitation deficits, the hydrological buffering capacity of below-ground components, namely ice-rich mountain permafrost and groundwater (Woo, 2011; Hayashi, 2020; Somers and McKenzie, 2020), is becoming increasingly important for water security and ecology (Millar et al., 2015; Reato et al., 2021). This is especially the case in water-stressed regions like semi-arid Central Asia (Karthe et al., 2015), the Himalayas (Jeelani et al., 2024), and the Dry Andes (Schaffer and MacDonell, 2022), but is also becoming increasing relevant for other rapidly changing mountain regions such as the European Alps.



Intact rock glaciers (RGs), common in many high-mountain ranges, are ice-rich permafrost landforms that store and release
water over various timescales ranging from seasonal to millennial (Jones et al., 2019). Their thick, seasonally-thawed active
layer (AL) insulates the ground and slows the melting of ground ice (Wakonigg, 1996; Humlum, 1997; Arenson et al., 2022;
Amschwand et al., 2024a). This renders permafrost ice more robust against climate change than glaciers on a long-term, multi-
decadal scale (Arenson et al., 2022; Haeberli et al., 2006). Three types of processes summarise the hydrological significance
of RGs. First, seasonal storage and freeze/thaw of water/ice in the AL act as a seasonal buffer by building ground ice in
autumn, winter, and spring (i.e., by trapping in-blown snow or refreezing snowmelt) and melting it during the thaw season.
Second, permafrost ice in the permafrost body (beneath the AL) is slowly released due to permafrost degradation as intact
(ice-bearing) RGs slowly transition towards a relict (ice-free) state. Since intact RGs are a common periglacial landform, the
amount of below-ground permafrost ice, as estimated from RG inventories and empirical area-volume scaling relations, is
substantial (Rangecroft et al., 2015; Azócar and Brenning, 2010; Brenning and Azócar, 2010; Brenning, 2005; Jones et al.,
2018b, a). Additionally, frozen talus slopes also contain ground ice (Mathys et al., 2024). In semi-arid, weakly glacierised
catchments, water equivalent (w.e.) volumes stored in RGs can exceed glacier ice volumes or may do so in the future (Jones
et al., 2018a; Bodin et al., 2010; Janke et al., 2017; Azócar and Brenning, 2010). Third, water is stored in the unfrozen fine-
grained sediments of intact and relict RGs and interacts while flowing through or beneath RGs (storage–release, routing and
chemical alteration/mineralization of water) (Azócar and Brenning, 2010; Corte, 1976; Burger et al., 1999; Jones et al., 2019).
The storage capacity of liquid water increases with progressing RG degradation, where the available pore space increases at
the expense of the lost ground ice (Wagner et al., 2016).

Ground ice volumes in the AL are ostensibly much smaller compared to those of permafrost ice; however, they are the
sites of rapid exchange between the atmosphere and hydrosphere. Marchenko et al. (2012) reported that observed seasonal
accumulation and melt of $40-60$ cm of ground ice stored a substantial amount ($< 30\%$) of the snowpack in the Northern
Tien Shan. Halla et al. (2021) found interannual ice storage changes on the Dos Lenguas rock glacier (Dry Andes of Ar-
gentina) of $-36$ mm yr$^{-1}$ ($25-80\%$ of the annual precipitation) and $+28$ mm yr$^{-1}$ ($17-55\%$). On the Murtèl RG, where our
present study is performed, seasonal ice storage changes in the AL were found to be $15-30$ cm w.e. ($10-32$ % of the annual
precipitation) based on point-scale below-ground stake measurements and AL energy budgets (Amschwand et al., 2024b).

Distributed, non-invasive, and portable near-surface geophysical methods are well suited for mountain permafrost monitor-
ing. Recent decades have seen the emergence of the subdomain of hydrogeophysics (Binley et al., 2015), wherein geophysical
techniques are developed and applied to address hydrological and hydrogeological questions, and cryogeophysics (Hauck and
Kneisel, 2008; Godio, A, 2019), to address cryosphere-related questions. As there is strong coupling of many hydrogeological
and cryosphere processes (van Tiel et al., 2024), hydrogeophysical and cryogeophysical methods and applications have con-
siderable overlap. Methods, such as electrical resistivity tomography (ERT), ground-penetrating radar (GPR), and seismics,
have found useful applications in alpine contexts. These include rockslide characterisation (e.g., Godio et al., 2006; Scapozza
et al., 2011; Cody et al., 2020), aquifer delimitation (e.g., McClymont et al., 2010; Christensen et al., 2020; Halloran et al.,
2023), glacier tomography (e.g., Egli et al., 2021; Dow et al., 2020; Church et al., 2019), and permafrost measurement (e.g.,
Maurer and Hauck, 2007; Draebing, 2016; Mewes et al., 2017). Geophysical methods can be considered as either stationary or





transient in their applications. Stationary measurements are suited to subsurface "mapping" and the differentiation of units or layers, while transient methods enable the measurement of spatio-temporal variations in water or ice content. The application of transient methods can reveal new, often quantitative, information on subsurface water and ice dynamics.

Gravimetry is a geophysical and geodetic (Jaramillo et al., 2024) set of techniques involving the measurement of $g$, acceleration due to gravity across the Earth's surface. The GRACE and GRACE-FO missions (Rodell and Reager, 2023) measure transient $g$ on a global scale, however the >200 km spatial "footprint" of space-borne gravimetry is of little use for field-scale investigations. Stationary (steady-state) gravimetry at the land-surface has a long history of providing information on geological interfaces. In permafrost studies, however, its use has been quite limited compared to other geophysical methods. Stationary gravimetry has been used to locate massive subsurface ice and ice-rich soils (Loranger et al., 2015) and to determine RG internal structure (Hausmann et al., 2007). At the Murtèl RG, a 1991 gravimetric survey was used to map the bedrock interface and spatial extent of layers observed in core samples (Vonder Mühll and Klingelé, 1994). Time-lapse gravimetry (TLG), sometimes referred to as 'differential gravimetry' or 'transient (micro)gravimetry', involves measuring temporal changes in $g$ at one or more locations. In alpine environments, use of the method is growing. McClymont et al. (2012) observed spatially-variable $\Delta g$ across a talus-moraine complex in the Lake O'Hara (Canada) headwater catchment during the snowfree period. Also over the snow-free period, Arnoux et al. (2020) recorded large decreases in $g$ in the Vallon de Réchy (Switzerland) relative to a point located outside the catchment. Both of these studies used the Scintrex CG-5, which has since been superseded by the Scintrex CG-6, a significantly more stable and accurate relative gravimeter (Francis, 2021). At the Zugspitze (Germany), Voigt et al. (2021) used a continuous time-series from a superconducting absolute gravimetry to investigate snowpack dynamics, while Gerlach et al. (2017) performed a similar investigation on the Vernagtferner glacier (Austria). TLG has not yet been explicitly used to measure permafrost or periglacial processes.

Here, we present the first TLG study of an active RG. At the well-documented Murtèl RG (e.g., Haeberli et al., 1998; Vonder Mühll et al., 2000; Hoelzle et al., 2002), we combine repeat measurements of gravity with repeat aerial imagery in order to determine the change in gravity attributable to AL melt over the thaw season. Additionally, we calculate and compare the 2024 Bouguer anomaly (BA) with those obtained in 1991 by Vonder Mühll and Klingelé (1994). We interpret these results in combination with point measurements (Amschwand et al., 2024c, b) to reveal spatial patterns in AL dynamics and multi-decadal permafrost loss. Finally, we discuss the potential of TLG for monitoring ice/water storage changes in the mountain cryosphere and give research perspectives on exploring permafrost–groundwater connectivity.

## 2  The Murtèl Rock Glacier

Murtèl RG (WGS 84: 46°25′47″N, 9°49′15″E; CH1903+/LV95: 2'783'080, 1'144'820; 2620–2700 m asl.; Fig. 1a), is located in a north-facing cirque in the Upper Engadine, a slightly continental, rain-shadowed high valley in the southeastern Swiss Alps (Fig. 1a). Mean annual air temperature (MAAT) is $-1.7°C$ and mean annual precipitation (MAP) is $\sim 900$ mm (Scherler et al., 2014). The RG covers an altitude range from 2620 (base of front) to 2720 m asl. (transition to talus) (Fig. 1b–d). The talus slopes ($2720-2800$ m asl.) connect the active RGs to the headwalls. The small (30 ha), non-glacierised Murtèl catchment





consists of permafrost-underlain, unvegetated debris (on rock glaciers and talus slopes) or bedrock. Murtèl is located at the lower permafrost margin and its forefield ($2600-2620$ m asl.) is permafrost-free (Schneider et al., 2012, 2013). The rock glacier sits in a bowl-shaped, glacially overdeepened bedrock depression that has been measured with a multi-geophysical approach combined with borehole logs (Vonder Mühll and Klingelé, 1994).

The lobate Murtèl rock glacier is ca. 300 m long, 180 m wide, and covered by a coarse blocky AL (debris mantle). Geophysical investigations revealed that its thickness varies according to the surface micro-topography, from $\sim 2$ m in the furrows to $\sim 5$ m beneath the ridges (Vonder Mühll and Klingelé, 1994; Vonder Mühll et al., 2000). Thus, the permafrost table shares the surface furrow-and-ridge micro-topography, albeit with attenuated relief. Fine material increases towards the AL base, but is overall sparse. The permeable coarse blocky AL appears to have a high permeability and very low water retention capacity (Springman et al., 2012). The Murtèl permafrost body between the seasonally thawed coarse blocky AL and bedrock (at $\sim 50$m) consists of three distinct layers (Vonder Mühll and Haeberli, 1990; Haeberli, 1990; Arenson et al., 2002): (1) massive ice, sparsely sand- and silt-bearing (3–28 m, supersaturated with over 90% ice content); (2) a layer of ice-saturated frozen sand (28–32 m) accommodating ca. 60% of the total/surface displacement (shear horizon); and (3) ice-saturated debris (32–50 m; 40% ice). Three boreholes, all located within 30 m distance, share this three-part stratigraphy, but also reveal small-scale material differences laterally (e.g., variable ice/sand content, lenses) and thermal anomalies suggesting non-diffusive heat transfer and intra-permafrost water flow (Vonder Mühll, 1992; Arenson et al., 2010). The extent and ice content of the ice-rich permafrost body is well known (Vonder Mühll and Haeberli, 1990; Vonder Mühll, 1992; Vonder Mühll and Holub, 1992; Vonder Mühll and Klingelé, 1994; Haeberli and Vonder Mühll, 1996; Vonder Mühll et al., 2000). Electrical resistivity tomography (ERT) and seismic refraction tomography data (Hauck, 2013; Arenson et al., 2010; Mollaret et al., 2019) indicate that the ice-rich permafrost core has an extent of approximately $150 \times 300$ m$^2$, amounting to a water volume equivalent (WVEQ) of $\sim 1.5 \times 10^6$ m$^3$.

## 3 Gravimetry

### 3.1 Basic principles of time-lapse gravimetry

Time-lapse gravimetry (TLG) is the measurement of acceleration due to gravity, $g$, at two or more points in time, usually at fixed locations, in order to determine $\Delta g(t)$. More precisely, TLG measures changes in the vertical component of gravity, which is affected by changes in the distribution of mass:

$$\Delta g(t) = G \iiint \boldsymbol{x} \cdot \hat{z} \frac{\Delta \rho(\boldsymbol{x}, t)}{|\boldsymbol{x}|^3} d^3 x \tag{1}$$

where $G$ is the universal gravitational constant, $6.674 \times 10^{-11}$ N m$^2$ kg$^{-2}$; $\hat{z}$, the vertical unit vector; and $\Delta \rho$, the change in density at location $\boldsymbol{x}$, relative to the measurement point. $\Delta g$ measurements are generally reported in units of mGal ($10^{-5}$ m s$^{-2}$) or $\mu$Gal ($10^{-8}$ m s$^{-2} \approx 10^{-9} g$). Individual surveys using a portable relative gravimeter generally include multiple points and, through repeat measurements at one or more reference stations, form one or more closed loops. After corrections to the raw data,



the residual effect on $g$ due to mass distribution changes is revealed. Local mass loss due to ice melt or decreased groundwater storage in the vicinity below a measurement location will result in a decrease in gravity between two repeat surveys.

## 3.2 Time-lapse gravimetry surveys

**Figure 1.** Locations of the gravity survey points and webcam photos from the survey dates. *a)* Locations of the Maloja and Silvaplana points. Inset: Location of study site within Switzerland. *b)* Locations of the local points on and in the vicinity of Murtèl RG. *c)* Webcam image (courtesy of PERMOS/University of Fribourg) of the RG during the first survey. *d)* Webcam image of the RG during the second survey. Both webcam images have approximate locations of points `MURTEL02-07` overlaid (`MURTEL01` and `08` not visible). Background maps from Swisstopo.

We carried out two microgravimetry surveys on 08.07.2024 and 11.09.2024 ($\Delta t = 65$ days) using a Scintrex CG-6 Autograv (Scintrex, 2018). A 60 s integration time was used, and, to ensure stability, the device was set to measure until three consecutive measurements within a tolerance of $\sim$2 $\mu$Gal were recorded. The average of this stable 180 s duration measurement period was used in the below-discussed analysis. To enable time-lapse measurements and to correct for instrument drift,





both surveys started and terminated at absolute gravimetric reference point no. 1018 (Swisstopo, 2024), a first-order point
($g = 980\,224\,831 \pm 3\,\mu\mathrm{Gal}$) located on a bedrock outcrop at the Maloja Pass. In addition to the `MALOJA` reference point, nine
points were surveyed during both surveys: Silvaplana church, 2 locations (`MURTEL01 & 08`) adjacent to the RG, and 6 loca-
tions on the RG (`MURTEL02-07`) (Fig 1). The total duration of each survey was <5h20m. A Leica GNSS DGPS system was
used to survey all points to $< 2\,\mathrm{cm}$ absolute accuracy. The on-rock glacier survey points were marked in spring to be revisited
in autumn. Additionally, the height of the gravimeter above the ground surface was manually measured at each location.

### 3.3 Gravity corrections

Several standard corrections were applied to the gravimetric data: tilt, internal temperature, height difference, Earth tides, and
drift. The first two, tilt and temperature, are automatically applied by the gravimeter to each individual measurement. Tilt was
maintained in the calibrated <10 arcsecond range during all measurements, and thus the internal corrections were $\lesssim 1\,\mu\mathrm{Gal}$.
The gravimeter was in a stable internal temperature state during the surveys, having being supplied with continuous power for
several days prior to the surveys, thus ensuring internal temperature corrections were within the calibrated range. Corrections
for height, Earth tides, and drift were all applied manually. During repeat measurements at a given location, the exact height
of the device can vary slightly due, in part, to offsets caused by levelling. Gravity decreases with elevation (free-air gravity
gradient):

$$\frac{\partial g}{\partial r} = -\frac{2GM_E}{r^3} \simeq -3.1\,\mu\mathrm{Gal\,cm^{-1}}, \tag{2}$$

where $M_E$ is the mass of the Earth and $r$ is the distance to its centre of mass. While the true value of the local vertical gravity
gradient (VGG) varies over Earth's surface, especially in complex terrain, $3.1\,\mu\mathrm{Gal\,cm^{-1}}$ is adequate for this correction of a
few mm.

Earth tides, whose magnitude can reach $\sim 300\,\mu\mathrm{Gal}$, are periodic variations in $g$ primarily due to the relative positions of the
sun and moon with respect to the Earth. Through tidal harmonic databases, Earth tides can be calculated to a level of precision
and accuracy that far exceeds gravimetric measurement accuracy. Modern gravimeters implement these corrections internally;
however, to ensure the highest possible accuracy, we chose to manually apply these corrections using the current state-of-the
art Kudryavtsev (2004) database, which contains over 28'000 wave groups. Through a *python* script, we used the `pygtide`
package (Rau, 2018), to calculate the Earth tides for each individual measurement to sub-nGal precision. These corrections,
which ranged from $-62.5$ to $+81.7\,\mu\mathrm{Gal}$ for the July survey and $-45.9$ to $+41.1\,\mu\mathrm{Gal}$ for the September survey, were applied
to the measured data. The resultant datasets were used for the calculation of, and correction for, instrumental drift, which
occurs in all relative gravimeters. The Scintrex CG-6 has low instrumental drift of $< 200\,\mu\mathrm{Gal\,day^{-1}}$ and $< 20\,\mu\mathrm{Gal\,day^{-1}}$
residual drift (Scintrex, 2018), with independent observations of $< 1.5\,\mu\mathrm{Gal\,h^{-1}}$ (Francis, 2021). By comparing the repeat
measurements at `MALOJA` (Fig. 1a), we calculated and corrected for total drift. Finally, the corrected data were referenced to
the `MALOJA` reference point to enable calculation of the temporal difference $\Delta g$ in the two surveys.





### 3.4 Bouguer anomaly (BA) calculations for comparison with 1991 gravimetric data

The gravimetric investigations of Vonder Mühll and Klingelé (1994), carried out in Summer 1991, provided information on the internal structure of the Murtèl RG. Raw data, such as the gravity station coordinates and non-terrain corrected gravity values, from the 1991 survey are unfortunately no longer available. Thus, while terrain corrections are not necessary for time-lapse gravimetry studies, since the data in the Vonder Mühll and Klingelé (1994) study are Bouguer anomalies (i.e., terrain corrected), we also perform these corrections to our gravimetric data to enable comparison. Furthermore, while our gravimetric stations `MURTEL05-06-03-07-08` correspond to "Profile 3" (Vonder Mühll and Klingelé, 1994; Vonder Mühll, 1993), we did not return to the exact locations of the 1991 stations.

Calculation of Bouguer anomalies ($g_{\mathrm{BA}}$) involves four corrections:

$$g_{\mathrm{BA}} = g_{\mathrm{measured}} - \left[\gamma_0 + c_{\mathrm{FA}} + c_{\mathrm{BP}} + c_{\mathrm{terrain}}\right] \tag{3}$$

where $\gamma_0$ is "normal" gravity as calculated from the ellipsoid, while $c_{\mathrm{FA}}$, $c_{\mathrm{BP}}$ and $c_{\mathrm{terrain}}$ are the free air, Bouguer plate and terrain corrections, respectively. While methods for all of these corrections (also sometimes referred to as 'reductions') are extensively documented (e.g., Nowell, 1999; LaFehr, 1991), there are several variations in the possible formulas and methods for each of these corrections. The methods reported by Vonder Mühll and Klingelé (1994), and also detailed in Vonder Mühll (1993), included an erroneous form of the 1967 International gravity formula in 2nd order expansion (Caputo and Pieri, 1968), a linear $(-0.3086\,\mathrm{mGal\,m^{-1}})\,z$ free air correction, and a linear *Bouguer* plate correction of $(4.19 \times 10^{-5}\,\mathrm{mGal\,m^{-2}\,kg^{-1}})\,\rho z$, where $\rho = 2670\,\mathrm{kg\,m^{-3}}$. Their reported terrain corrections included two inner Nagy (1966) prism zones and two outer line-of-mass zones to 167 km. Our application of these methods did not result in BAs in their reported $-157.0$ to $-155.4\,\mathrm{mGal}$ range. Such issues related to reproducing historic BAs published in the Swiss Gravimetric Atlas (Swisstopo, 1994) have been encountered and documented by others (Bandou et al., 2022). *Swisstopo*, the Swiss Federal Office of Topography, has also confirmed these issues related to incomplete documentation of the processing of historic datasets. We thus apply modern standard corrections in order to enable comparisons of BAs relative to others in the same survey. Specifically, we use the full GRS80 Somigliana formula (Moritz, 1980) for normal gravity, the Wenzel free-air correction (Wenzel, 1996) with ETRS89 latitudes, and the Talwani Bouguer plate formula (Talwani, 1973) with a density of $2670\,\mathrm{kg\,m^{-3}}$. Terrain corrections were performed using the *swisstopo* `GRAVNIV` software with the publicly available 0.5m Swiss digital terrain model `swissALTI3D`, as discussed in Bandou et al. (2022) and further detailed in Appendix 3A of Bandou (2023).

### 3.5 Digitization of the 1991 survey data

As discussed in Sect. 3.4, the only sources of the 1991 survey data currently available are the graphical data in Vonder Mühll and Klingelé (1994) and Vonder Mühll (1993) (both of which contain the same information). We used a digitization tool to extract, directly from the figures, the positions of the 1991 gravimetry survey points and the values of the calculated BAs and relative position along "Profile 3" data. Accuracy in the BA data digitization procedure can be considered as $\pm \sim 10\,\mathrm{\mu Gal}$ and $\sim 2\,\mathrm{m}$ along the profile. Planar positions can be considered accurate to $\pm \sim 3\,\mathrm{m}$, although we note some discrepancy between



planar positions on the original map (Vonder Mühll, 1993) and the distances along the original "Profile 3". In the absence of other information, we present the 1991 survey data as originally published.

## 4    Snow spatial distribution and corrections using aerial imagery

In order to distinguish between surface and subsurface cryosphere processes in our gravimetric analyses, we need to account for the effect of snow present during the first survey. To accomplish this, we used photogrammetry with UAV imagery.

### 4.1    Aerial imagery data acquisition

On both survey dates, we obtained aerial images using a compact DJI Mini Pro 4 with a 48 megapixel camera. The drone was piloted over the lower part of the RG to ensure coverage across the target zone. Due to the presence of overhead aerial cables and an operating gondola lift, extended zones were not surveyed. In total, 375 images were acquired during the July 2024 survey and 268 images during the September 2024 survey. As non-RTK UAVs have poor positional accuracy, we also established and measured ground control points (GCPs) in order to improve outputs from the photogrammetry processes. We used a Leica GNSS RTK surveying system to measure the 3-D positions of eight GCPs, colocated at the gravimetric survey points `MURTEL01-08` (Fig. 1), to $0.1\,\mathrm{cm}$ precision and $< 2\,\mathrm{cm}$ accuracy.

### 4.2    Photogrammetry

Digital surface models (DSMs) and orthomosaics were created for both survey dates. We used the photogrammetry software Pix4Dmapper (v4.9.0, 2023, Pix4D S.A., Switzerland). The average ground sampling distance was 2.04 cm for the July survey and 3.45 cm for the September survey. A high number of median keypoints per image ($>72000$ for both surveys) were determined, each with a minimum of four matches across multiple images. We manually georeferenced six GCPs (`MURTEL02-07`) that could be located in multiple images with high precision and accuracy. The georeferenced point clouds were used to produce DSMs and orthomosiacs with 5 cm resolution. We used noise filtering and inverse distance weighting, but no surface smoothing, for the DSM processing. Comparison of the final results revealed only very small offsets at the sub-pixel scale.

### 4.3    Snow density measurements

We measured the snow density using a portable digital balance (precision 0.1 g) and a coring tube (inner diameter 5 cm). The volume and mass of eighteen samples were measured. The measured sample sizes ranged from $137-736\,\mathrm{cm}^3$ and from $76-410\,\mathrm{g}$. Measured densities ranged from $504-587\,\mathrm{kg\,m}^{-3}$, a typical range for névé (Fierz, C. et al., 2009). The average snow density was determined to be $550 \pm 27\,\mathrm{kg\,m}^{-3}$.





## 4.4 Calculation of snow gravity effect

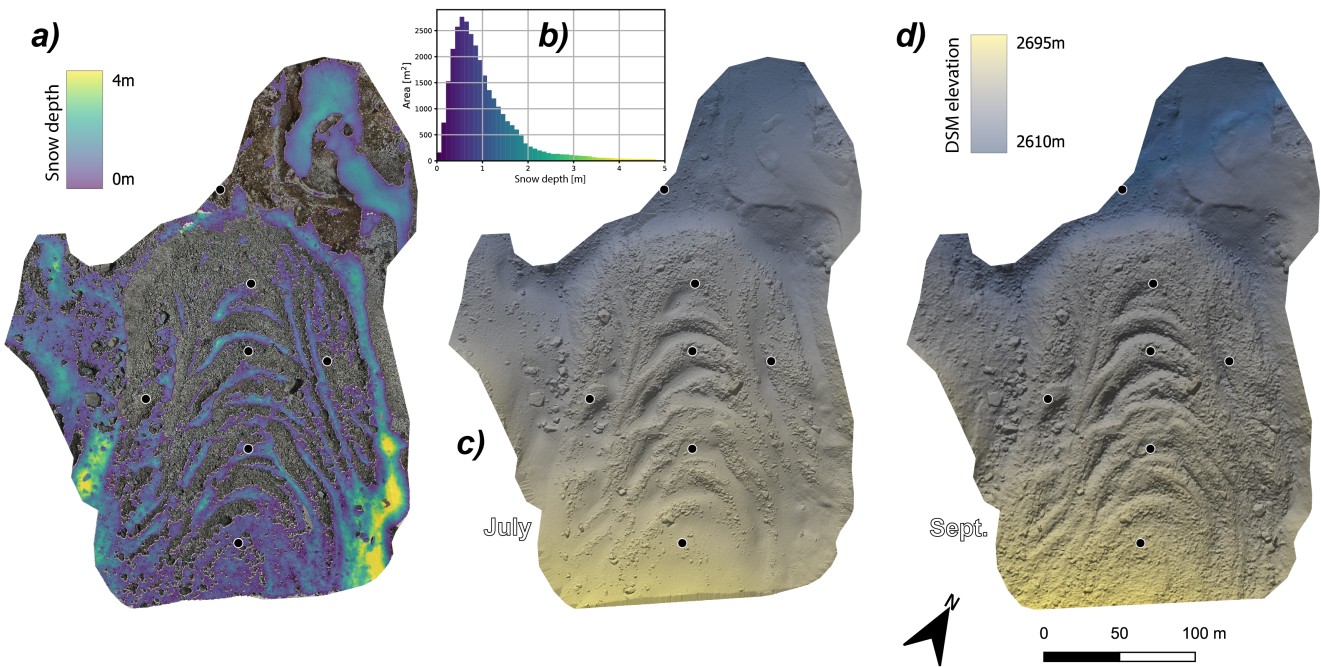

**Figure 2.** Snow depth and the digital surface models (DSM) calculated via photogrammetry and manual fixed point referencing. *a)* The depth of the snow present during the July 2024 survey and absent in the Sept. 2024 survey. *inset b)* A histogram of area covered by the snow with depths in 10 cm intervals. *c)* The DSM for the 08.07.2024 survey. *d)* The DSM for the 11.09.2024 survey. Only areas with sufficient coverage by both surveys are shown. The black dots indicate the locations of our gravimetric survey points (Fig. 1).

The gravitational effect of the snow, present in the July survey and absent in the September one, at each of the survey points was calculated through a multistep process: 1) creation of a snow raster mask based on the July orthomosaic, 2) calculation of the height difference between the DSMs, 3) fusion of these data to create a 3D differential snow volume model, and 4) forward calculation of the vertical gravity effect at the gravity survey points. *Python*, implementing standard packages (*numpy*, *matplotlib*, etc.) and *rasterio*, was used for these processes. Throughout, manually clipped DSMs and orthomosiacs were used,
limiting the analysis to snow-covered regions with high quality DSM data for both surveys.

We first created a spatial mask for the snow-covered pixels using the July 2024 orthomosiac (Sect. 4.2). This georeferenced image was converted to a luminance ($L$) greyscale format as per the Rec. 709 fomula:

$$L = 0.2125R + 0.7154G + 0.0721B \tag{4}$$

where $R$, $G$, and $B$ are the red, green and blue pixel intensities, respectively. The luminance image was blurred using a uniform,
centred filter of dimensions 1.2×1.2 m (25×25 pixels). This was necessary in order to avoid isolated high luminosity pixels being identified as snow. A simple 55% threshold was applied to the processed image, with the exception of the low-albedo





snow in the southeast corner of the surveyed zone, where a 38% threshold was used. This process resulted in satisfactory identification of snow-covered regions, while avoiding misidentification of non-snow covered regions such as high-luminosity boulders. The height difference between the July and September DSMs was evaluated by a simple difference between the two.

The above discussed snow mask, as well as a 5 cm thickness lower bound, were applied to this raster data. The final result was a 2-D map of snow thickness (Fig. 2), which, when combined with either the July or September DSM, provides 3-D information on the change in snow volume distribution.

To calculate the decrease in gravity due to the melting snow, we performed forward gravity modelling using a modified version of Gravi4GW (Halloran, 2022) that implements the Nagy prism (Nagy, 1966) rather than equivalent point masses. A

correction for the height of the proof mass (6.58 cm above the bottom of the gravimeter, Scintrex, 2018) relative to our surveyed coordinates (Fig. 1b) was also applied, although the effect was minor for all points ($0.0-0.3\,\mu\mathrm{Gal}$).

Additionally, when comparing the digital surface models from the two surveys, we discovered a new, large boulder, ostensibly originating from the unstable zone located adjacent and above (southwest) the RG. We estimated the boulder's mass to be $< 1 \times 10^5$ kg. The nearest gravimetry point, MURTEL04, was located $\sim 25$ m away and $\sim 1$ m lower than the boulder's centre

of mass. The straight-line gravitational pull of this boulder on MURTEL04 was thus $\lesssim 1\,\mu\mathrm{Gal}$ and the vertical component (i.e., that which we measure in time-lapse gravimetry) $< 0.1\,\mu\mathrm{Gal}$. Thus, we conclude that this new rock mass can be ignored in our analyses.

## 5 Results

### 5.1 Seasonal gravity variations

In order to correct for the gravity effect of the snow present in the July 2024 survey, the net gravitational effect on the survey points MURTEL01-08 was calculated (Sect. 4.4). This vertical $\Delta g$ effect of the snow ranged from $-0.58$ (MURTEL04) to $-11.39\,\mu\mathrm{Gal}$ (MURTEL06) (Table 1). Only the vertical component of snow-induced gravity change has an effect on measured gravity (Eq. 1 and Sect. 6). While a significant amount of snow was present (Fig. 2), the majority of its gravitational effect is non-vertical. For reference, 1 cubic meter of snow (with a density of $500\,\mathrm{kg\,m^{-3}}$) at a horizontal distance of 10 m, centred 1 m

below a given point, would have a vertical gravity effect of $\sim 3.6 \times 10^{-3}\,\mu\mathrm{Gal}$. We also note that while snow-covered regions extending hundreds of metres from the RG could not be surveyed, these zones are generally at higher elevations than the survey points. The gravitational effect of the snow in those lateral zones would thus be small and positive (McClymont et al., 2012). The limited spatial coverage of the DSMs would thus have the effect of underestimating net mass loss in the AL and subsurface water. Finally, the statistical uncertainty in the measured snow density was 4.9%, which translates to an uncertainty ranging

from $\pm 0.03$ (MURTEL04) to $\pm 0.56\,\mu\mathrm{Gal}$ (MURTEL06).

Accounting for the snow, net gravity changes were negative at all eight survey points (Table 1), indicating a local decrease in subsurface mass. With the exception of MURTEL04, located near the western flank of the RG, all net gravity changes in the approximate range of $-5$ to $-15\,\mu\mathrm{Gal}$. Along the central profile, $\Delta g$ decreases in magnitude from $-15.2\,\mu\mathrm{Gal}$ near the lower rooting zone to $-5.2\,\mu\mathrm{Gal}$ near the front. MURTEL08, located on bedrock $\sim 20$ m from the RG front, experienced a decrease



| Point name | Processed $\Delta g$ ($\mu$Gal) | Snow effect $\Delta g_{snow}$ ($\mu$Gal) | $\Delta g$, after snow effect correction ($\mu$Gal) |
|---|---|---|---|
| MURTEL01 | −11.7 | −2.7 | −9.0 |
| MURTEL02 | −13.7 | −3.3 | −10.3 |
| MURTEL03 | −11.9 | −6.0 | −5.9 |
| MURTEL04 | −29.8 | −0.6* | −29.2 |
| MURTEL05 | −19.3 | −4.0 | −15.2 |
| MURTEL06 | −22.7 | −11.4 | −11.4 |
| MURTEL07 | −10.3 | −5.2 | −5.1 |
| MURTEL08 | −11.4 | −0.8* | −10.6 |

*Incomplete DSM coverage of surrounding terrain.

**Table 1.** $\Delta g$ results after processing (Sect. 3.3) and $\Delta g - \Delta g_{snow}$ after accounting for the effect of surface snow mass $\Delta g_{snow}$.

in gravity of $-10.6\,\mu$Gal. Results from the two lateral survey locations, MURTEL02 near the east flank ($-10.3\,\mu$Gal) and MURTEL04 to the west ($-29.2\,\mu$Gal) differ significantly, although limited DSM coverage may have a minor influence on the snow-corrected value at MURTEL04. Overall, while there is a spatial trend along the central line, the results point to significant heterogeneity in the subsurface processes. For first-order interpretation, the *Bouguer* plate approximation (BPA), an infinite flat plane where a decrease of $2.38\,$cm in water level equivalent equates to a $-1\,\mu$Gal change, provides a rough starting point,
implying decreases in subsurface water storage from $12-70\,$cm of water equivalent (see Sect. 6.1.1 for an in-depth discussion and quantitative interpretation).

For uncertainty estimation, we follow the same logic as previous TLG investigations (McClymont et al., 2012; Arnoux et al., 2020), noting that these two studies used Scintrex CG-5 devices, which have significantly worse temperature and tilt stability and greater drift rates compared to the CG-6 device used in our study (Francis, 2021). In terms of uncertainty, we assume three
significant and uncorrelated sources of uncertainty, in addition to the afore discussed uncertainty in the snow and rock fall correction (Table 2): precision, drift, and elevation. Precision groups together uncertainty related to vibrations, tilt corrections and internal temperature corrections (Sect. 3.3). As determined from standard deviation of the three repeats whose average was used to determine the final values, we observe values ranging from $\pm 0.2$ to $\pm 2.4\,\mu$Gal with an average of $\pm 0.9\,\mu$Gal for the July survey and $\pm 0.05$ to $\pm 1.1\,\mu$Gal with an average of $\pm 0.6\,\mu$Gal for the September survey. Adding these uncertainties
together in quadrature for each survey point we obtain precision uncertainties in $\Delta g$ of $\pm 0.7$ to $\pm 2.5\,\mu$Gal. We assume an uncertainty associated with the impact of non-linear drift of $\pm 4.4\,\mu$Gal based on survey duration and device specifications. Finally, uncertainty in $\Delta g$ associated with uncertainty in the height of the device is assumed to be $\pm 1\,\mu$Gal as we manually measured the height of the device to $\lesssim 0.3\,$cm precision (Eq. 2) in addition to surveying. Calculations of these uncertainties,





combined with the snow and rock fall correction uncertainty for individual points, range between $\pm4.6$ to $\pm5.2\,\mu$Gal and thus

we arrive at a total estimated uncertainty in the snow cover-corrected $\Delta g$ of $\pm5\,\mu$Gal.

| Source | Value ($\mu$Gal) |
|---|---|
| Instrument precision | $\pm0.7$ to $\pm2.5$ |
| Instrument drift | $\pm4.4$ |
| Elevation differences | $\pm1$ |
| Snow | $\pm0.03$ to $\pm0.6$ |
| Rock fall | $< \pm0.1$ |
| Total* | $\pm5$ |

*The total uncertainty is calculated from the square root of the
sum of the squares of the individual uncertainty components.

**Table 2.** $\Delta g$ uncertainty budget.





## 5.2 Comparison of the 1991 and 2024 gravity surveys



**Figure 3.** *a)* Bouguer anomalies (BAs) and *b)* approximate relative positions. The smoothed trend line (Sect. 3.5) from the 1991 survey (Vonder Mühll and Klingelé, 1994; Vonder Mühll, 1993) and those calculated from our 2024 surveys (Sect. 3.4) are shown. Distance refers to that of "Profile 3" in the 1991 survey, starting from its uppermost point. Approximate distances of the 2024 survey points in proximity to the 1991 profile are shown. An additional point near "Profile 1" in the 1991 survey is not shown: −152.447 and −152.476 mGal in July/Sept. 2024 vs. ∼ −156.15 mGal in 1991 (Vonder Mühll and Klingelé, 1994). Note that due to lack of documentation and differences in methodologies (Sect. 3.4), the values cannot be compared directly, although spatial trends and the relative values within each survey can.



As discussed (Sects. 3.4–3.5), even though most TLG applications do not require normal gravity, free air, *Bouguer* plate, and terrain corrections, we needed to calculate and apply these to calculate the *Bouguer* anomaly for comparison with the digitised 1991 survey data. Furthermore, due to incomplete documentation of the processing steps in Vonder Mühll and Klingelé (1994)

and lack of the original DSM files, absolute differences between the published 1991 values of our BA values differ by an offset which cannot be known exactly (Sect. 3.4). While uncertainty of the 1991 survey data is not extensively discussed in Vonder Mühll and Klingelé (1994) or Vonder Mühll (1993), the techniques and technology of the time clearly limited the achievable accuracy of the data. The LaCoste and Romberg Model G gravimeter used in 1991 had a repeatability of $100 \, \mu$Gal and accuracy $< 40 \, \mu$Gal. The higher density of points in that study (1 every $\sim 25 \, \mathrm{m}$ in the central line profile) allowed for the

calculation of a smoothed trend line which we use for comparisons on a relative basis (Fig. 3).

By displaying the calculated BAs with an offset of $3.9 \, \mathrm{mGal}$ (Fig. 3), we can approximately match our off-RG forefront survey point, MURTEL08 (Fig. 1), to the 1991 smoothed line in order to facilitate visual comparisons of the relative values. Our Bouguer anomalies display a larger difference between the forefront region (MURTEL08) and front and central RG zones (MURTEL07, 03, 06) than in the 1991 survey. In contrast, the difference between the forefront and lower rooting zone at the

transition to the talus slope (MURTEL05) is similar to that observed in the 1991 survey. Comparing with the smoothed historic data, the relative difference between the forefront and the front/central zones appears approximately $110-190 \, \mu$Gal greater in 2024 than in 1991. Similarly, comparing MURTEL08 to MURTEL04 (Fig. 3), the difference between the BAs at these points increases by $\sim 200 \, \mu$Gal between surveys. Our other survey points, MURTEL01 and 02, do not overlap with any of the 1991 survey points. Their BA values are $-152.241 \, \mu$Gal (July) $\&$ $-152.253 \, \mathrm{mGal}$ (Sept.) and $-152.550$ (July) $\&$ $-152.564 \, \mathrm{mGal}$

(Sept.), respectively. Given the differences in the locations of our survey points, caution should be taken in drawing quantitative conclusions based on this comparison.

# 6 Discussion

## 6.1 Seasonal processes and gravity

### 6.1.1 Seasonal storage changes in the active layer

The corrected July–September $\Delta g$ values (Table 1 & Fig. 4) result from the integrated effect of mass distribution changes surrounding the survey point, weighted by distance $r^{-2} \cdot \hat{z}$ (Eq. 1). Corrections for change of mass on the surface, in the form of snow (Sect. 4), allow for local subsurface processes to be isolated. The net negative gravity changes $\Delta g$ indicate a decrease in total water storage (ground ice and liquid water)–that is, net export of ice/water out of the rock glacier. While uncertainties exist (see discussion below), we attribute the dominant $\Delta g$ signal over the thaw season to ice storage changes in the AL. Our

reasoning is based on previous observations, specifically below-ground stake measurements of the moving ground-ice table and ground ice melt inferred from calculated AL energy budgets (Amschwand et al., 2024b). In order to estimate the water equivalent (w.e.) of this change in ice mass, we employ the *python* tool GRAVI4GW (Halloran, 2022), which takes into account the non-horizontal and non-planar topology of a changing unconfined groundwater table or, in our case, that of the ground-ice



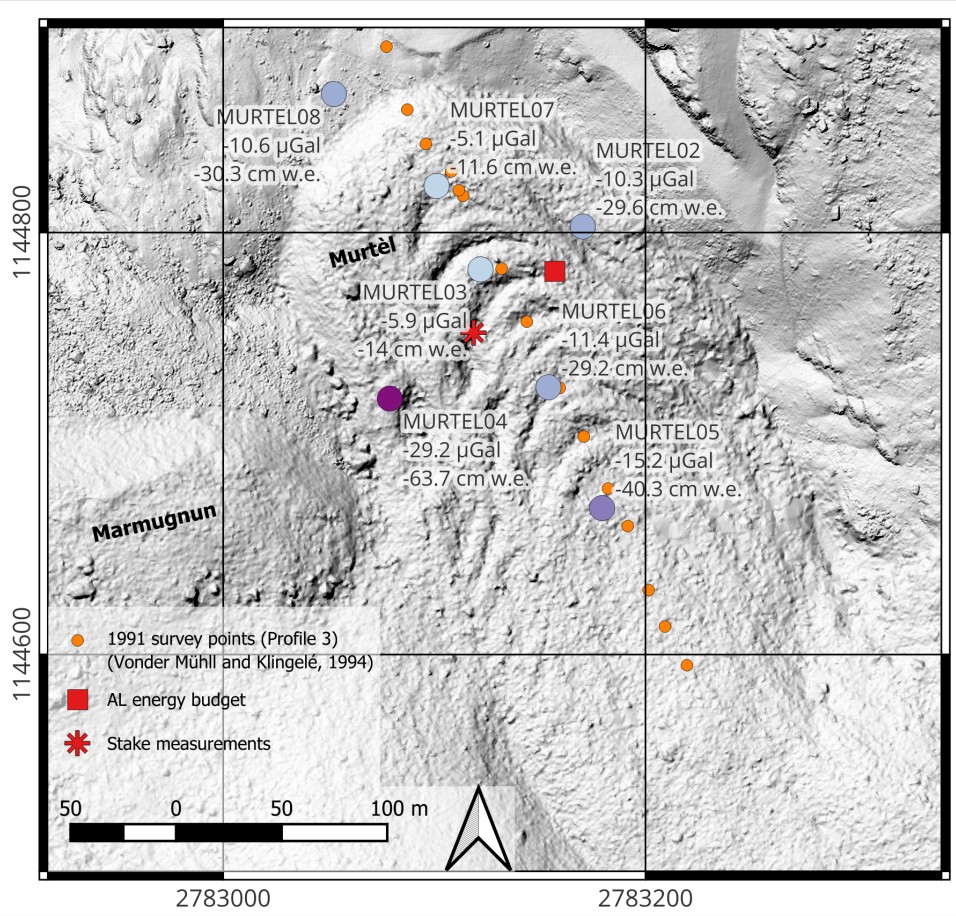

**Figure 4.** 2024 seasonal gravity and total water storage changes at survey points MURTEL02-08. Total water storage (in cm w.e.) calculated with $h_{\text{eff}} = 4$ m (uncertainty ranges shown in Fig. 5).

table (AL base). The sensitivity of gravity changes $\Delta g$ to total water storage changes,

$$\beta_z(\boldsymbol{x}, h_{\text{eff}}) = \frac{\partial \boldsymbol{g}}{\partial h_{\text{eff}}} \cdot \hat{z} \tag{5}$$

varies with the effective depth of the groundwater/ice table $h_{\text{eff}}$ and is influenced by the surrounding topography (Halloran, 2022). $\beta_z$ varies with terrain slope and curvature just as the local vertical gravity gradient (VGG) does (Zahorec et al., 2024) and is highest on convex topographic features (mounds, ridges). A related concept is topographic admittance (e.g., Chaffaut et al., 2022; Voigt et al., 2021), which is most relevant for thin surface water accumulation (e.g., in the form of snow). The influence of small-scale topographic features decreases, while the effective radius of influence or "footprint" increases, with increasing



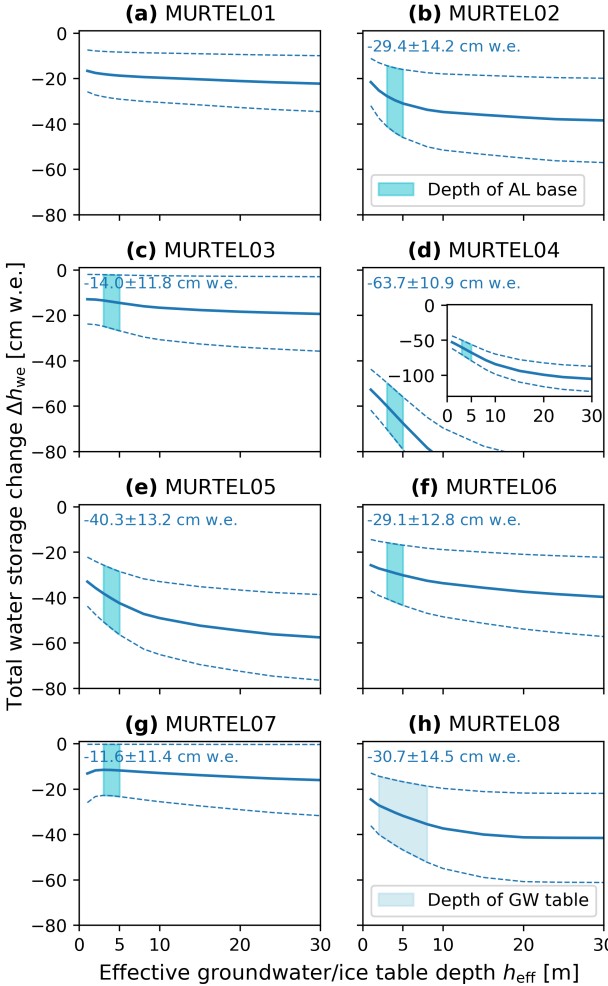

**Figure 5.** Calculated seasonal below-ground total water storage change $\Delta h_{we}$ for a range of hypothetical $h_{\text{eff}}$ and given $\Delta g \pm 5\,\mu$Gal for the on-RG survey points `MURTEL02-07` (Fig. 4 for locations of on-RG survey points). The off-RG survey points `MURTEL01 & 08` are included for completeness.

$h_{\text{eff}}$ (Halloran, 2022). Based on field observations (Amschwand et al., 2024b), PERMOS borehole temperature series (Noetzli and Pellet, 2023), and geophysical investigations (Vonder Mühll and Klingelé, 1994; Vonder Mühll et al., 2000; Mollaret et al., 2020)), the AL depth can be assumed to be in the range of $2-5$ m. Using this range of values for $h_{\text{eff}}$, the measured $\Delta g$ values (with the $\pm 5\,\mu$Gal uncertainty) translate to an estimated water equivalent of AL melt $\Delta h_{we}$ between surveys in the range of 330 $15-40$ cm w.e. for all on-RG locations except `MURTEL04` (Figs. 4, 5).

TLG measurements are fundamentally ambiguous because the gravitational effect depends on the magnitude, distance, and relative position of the mass change relative to the survey point (Eq. 1). Two features of TLG are noteworthy (and discussed further in Sect. 6.3): Gravity does not discriminate between water in its solid or liquid state, and is in places weakly sensitive to




the depth where the water storage changes occur (Chaffaut et al., 2022; Halloran, 2022). As a consequence, different ground-water and ice storage changes– whether localized and shallow or extended and deep (or any combinations thereof)–can produce the same $\Delta g$ signal that reflects the total water storage change below ground (i.e., excluding snow). This is shown in Fig. 5, where total water storage change $\Delta h_{we}$ is plotted as a function of depth $h_{\mathrm{eff}}$ to the (hypothetical) water/ice table. While other subsurface processes (Sec. 6.1.2) may play some role, the inferred $\Delta h_{we}$ of $15-40\,\mathrm{cm}$ w.e. compares well with the results of Amschwand et al. (2024b) ($15-30\,\mathrm{cm}$ w.e.), suggesting that the snow-corrected on-RG $\Delta g$ are dominantly a signal of ice storage changes in the AL. This interpretation is most convincing at the mid-frontal survey points MURTEL06, MURTEL03, and MURTEL07 (Fig. 4), which are in the vicinity of direct AL thickness measurements (Amschwand et al., 2024b). Under the assumption that the dominant mass redistribution effect stems from AL melt, the uncertainty in AL storage change water equivalent, $\Delta h_{we}$, is in the range $10-14\,\mathrm{cm}$ w.e. for survey points on the RG. This uncertainty stems from both the uncertainty of the final, snow-corrected $\Delta g$ values and the uncertainty in the local AL depth below the surface. The sensitivity–depth relation $\beta_z$ (Eq. 5) differs between survey points and varies with terrain slope and curvature. The $\Delta h_{we}$ uncertainty stemming from $h_{\mathrm{eff}}$ (as opposed to from $\Delta g$ measurements) is influenced by topography and can be calculated beforehand to optimize the survey point locations. Survey points on moderately convex features (ridges and plateaus) generally give higher sensitivity to storage changes, $\beta_z$, that varies little with effective depth $h_{\mathrm{eff}}$. This is illustrated by the mid-frontal survey points MURTEL06, MURTEL03, and MURTEL07, where $\beta_z$ (and consequently $h_{we}$) varies little with $h_{\mathrm{eff}}$. There, the $\Delta h_{we}$ uncertainty stems mainly from $\Delta g$ uncertainty (Table 2) rather than that of $h_{\mathrm{eff}}$. With a $\beta_z$ (Eqn. 5) weakly sensitive to depth $h_{\mathrm{eff}}$, total storage change estimates $h_{we}$ are only weakly impacted by the uncertainty in AL depth or the possible occurrence of deeper thaw processes.

### 6.1.2 Other seasonal-scale processes

In principle, $\Delta g$ is sensitive to more than just ice loss by AL thaw. Other changes in total water storage, in both solid and liquid states, and mass movements can also affect gravity at the RG surface (Fig. 6). These include effects from snow, intra-permafrost melt and subsidence, groundwater export, and rock movements. Snowpack can significantly contribute to the $\Delta g$ signal and must be accounted for by calculating and subtracting its differential gravitational effect (Sect. 4). Based on the drone-derived differential DSMs and the snow density measurements, the snow cover effect could be precisely calculated (within $\pm 0.6\,\mu\mathrm{Gal}$, Table 2) and amounted to $\sim 20-50\%$ of the measured $\Delta g$ signal (Table 3), except at survey point MURTEL04. There, near the western rock glacier margin, the snow's vertical gravity effect was counter-intuitively weak despite large snow depths locally exceeding $4\,\mathrm{m}$ in the nearby depressions (Fig. 2). The small gravitational effect of the snow occurred here because mass loss above and below the survey point have opposing effects on the measured vertical $\Delta g$ and have, effectively, partially cancelled each other out (melt at higher elevation increases $g$, while melt at lower elevation decreases $g$). However, limited extended DSM coverage towards the adjacent Marmugnun rock glacier precludes a conclusive uncertainty analysis for the correction at this point.

Below-ground water/ice storage changes (in excess of the pore space) can result in subsidence or heave. At Murtèl, however, subsidence (loss of excess ice) is $\leq 1\,\mathrm{cm}\,\mathrm{yr}^{-1}$, an order of magnitude smaller than the seasonal ice storage change (Am-



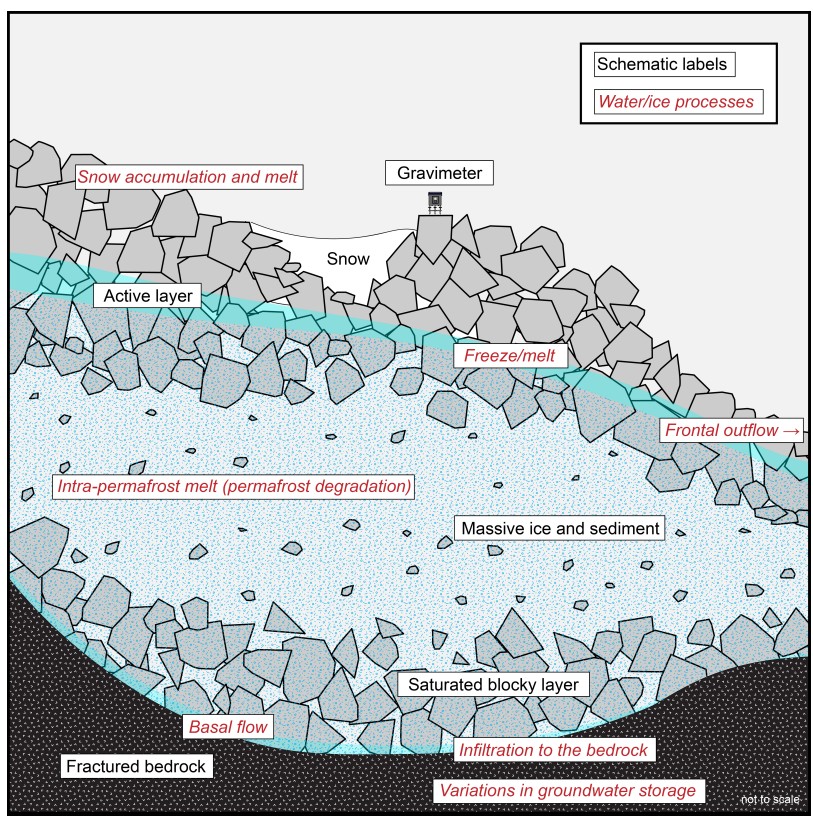

**Figure 6.** Conceptual longitudinal cross-section model (not to scale) displaying water/ice storage changes that occur in rock glaciers. While the structure and bowl-shaped base of Murtèl are integrated here, the displayed processes may occur in all rock glaciers. On the seasonal scale of our July-September 2024 TLG measurements, superficial melt of snow (which we account for in Sect. 4.4) and near-surface AL thaw (discussed in Sect. 6.1.1) are the dominant processes. On multi-year timescales, all of the processes (labelled in red), as well as rock mass movements (Sect. 6.2), may play a role. These processes may alter the value of $g$ as measured at the surface.

schwand et al., 2024b). While the presence of liquid water in the permafrost core of Murtèl (and other rock glaciers) is well documented from borehole drilling (Vonder Mühll and Haeberli, 1990; Arenson et al., 2002; Bast et al., 2024), little is known

370 about temporal changes in intra-permafrost water content. Based on the weak heat fluxes (Amschwand et al., 2024c) and the low hydraulic permeability of the ice-rich permafrost body (Springman et al., 2012), we consider effects of intra-permafrost thaw (that would decrease $g$) and changes in water content negligible on seasonal timescale compared to the substantial ice turnover in the AL.

Groundwater export from the base of the Murtèl RG, lying in an over-deepened bowl-shaped depression (Barsch and Hell,

375 1975; Vonder Mühll and Klingelé, 1994), is expected to be rather limited due to the presumably low hydraulic conductivity of the crystalline bedrock (granodiorite). Based on the observation that the RG springs dry out in hot–dry summer periods when melt rates are highest, Amschwand et al. (2024b) found that a significant aproportion of Murtèl meltwater does not leave the



catchment as surface streamflow and thus most likely recharges the sub-permafrost aquifer. Halla et al. (2021) reached the same conclusion for a rock glacier in the Dry Andes. The influence of groundwater recharge at the RG base is nonetheless impossible to verify without bedrock-intersecting piezometric levels, hydraulic characterisation of the fractured bedrock, or detailed water balance measurements and calculations. This cryosphere-groundwater connectivity, whose relevance may be significant on longer timescales, is further discussed in Sect. 6.4.

As for non water/ice mass redistribution phenomena, accumulation of rock mass above and near the survey points could effectively reduce $g$. Due to the inverse-distance-squared weighting (Eq. 1), mass movement effects on $\Delta g$ would be the most pronounced at survey points closest to the active root zone (talus slope) and to the nearby Marmugnun RG (MURTEL04), although this depends on the individual rockfall trajectories. The gravitational pull of a large rock-fall boulder deposited between the two time-lapse survey dates has been calculated from the two DSMs (Sect. 4.4) and these processes are deemed insignificant at seasonal scale, although this depends on the exact locations of survey points and rockfall trajectories. In this study, we do not account for processes occurring in neighbouring regions, although we note that decreasing mass at elevations above our survey points would be positive and thus have the effect of "masking" the change in gravity imparted by AL melt and other subsurface ice and water mass loss in the RG.

## 6.2   Multi-decadal processes and gravity

Comparison of the 1991 survey data (Vonder Mühll and Klingelé, 1994) and our measurements must be nuanced by the technological and methodological factors discussed in Sect. 5.2. Nevertheless, some interpretations and hypotheses can be set forth. While over a single thaw season, or part thereof, melt of ground ice in the AL is the dominant mass redistribution process, over multi-year timescales–in this case, 33 years–there are several processes that may significantly affect $g$.

The starkest change in the spatial trends of the BA (Fig. 3) is the relative decrease in gravity in the central and front region (MURTEL06, 03 & 07) as compared to both the forefront measurement point (MURTEL08) and that in the lower rooting zone (MURTEL05). Although MURTEL08 is located $\sim 25\,\mathrm{m}$ from the 1991 survey profile, the others are all located $< 5\,\mathrm{m}$ from it. Thus, even if we ignore the forefront point, a clear difference in trend can be observed where the BA is now lower relative to that of the rooting zone than it was in 1991. Mountain permafrost regions are highly dynamic environments characterized by mass moving processes (rock/debris) and total water (water, snow, ice) storage changes acting on different magnitudes, rates/timescales, and depths below ground, i.e., on the surface, in the AL (supra-permafrost), in the permafrost body (intra-permafrost), and beneath (sub-permafrost). The Murtèl periglacial cirque is typical in this respect (Müller et al., 2014). Here, we discuss processes related to both rock mass movements and water/ice storage changes that could be relevant for decadal changes in gravity in alpine periglacial settings (Table 3).

Landform and rock mass movements involve gradual or episodic movement of mass. Active rock glaciers undergo creep at rates of a few $\mathrm{cm\,yr^{-1}}$ to $\mathrm{m\,yr^{-1}}$ (Delaloye et al., 2010; Noetzli and Pellet, 2023; Manchado et al., 2024; Louis et al., 2024) and destabilized rock glaciers can move at rates in the $\sim \mathrm{cm\,day^{-1}}$ range(Marcer et al., 2021; Hartl et al., 2023). Even at slowly deforming RGs, such as Murtèl, the effects of creep on mass distribution, and thus, potentially $g$ will be significant on multi-year timescales. Furthermore, RG creep will eventually render it impossible to return to the same survey points for



repeat surveys as the point in space may end up being within the rock mass or high above the land surface. Although Murtèl exhibits comparatively slow creep rates of $\lesssim 10\,\mathrm{cm\,yr^{-1}}$ amounting to $\sim 2{-}4\,\mathrm{m}$ of horizontal displacement over 33 years, the displacement alone is very unlikely to account for the 1991–2024 differences. We note that shifting these 2024 BA points by
$\sim -30\,\mathrm{m}$ along x-axis in Fig. 3 would result in an overlap with the 1991 trend line. Rock fall and other landslide-type events also redistribute mass. The interplay of the effects of rock debris from headwalls above rooting zones and those of RG creep is highly complex and would require careful and extended topographic surveying (i.e., with LiDAR or photogrammetry) at the time of each gravimetric survey. When conceptualising the gravitational effects of these, and, indeed, all mass redistribution processes, it is important to remember the $r^{-2} \cdot \hat{z}$ weighting (Eq. 1) of mass changes. All things being unchanged, if mass is
removed from an elevation above the point and added at an elevation below a point, then gravity will increase. However, the effect of falling rock mass is dependent on the relative distance and vertical orientation relative to a measurement location, both before and after the rockslide episode. Finally, tectonic uplift may be important to consider for multi-decadal TLG surveys, although its effect would be uniform across such a small study area at the Murtèl RG. In the Bernina Alps, tectonic uplift rates are about $1{-}2\,\mathrm{mm\,yr^{-1}}$ (Sternai et al., 2019), thus tectonic uplift contributes not more than $-0.6\,\mathrm{\mu Gal\,yr^{-1}}$ (Eq. 2) or
$-20\,\mathrm{\mu Gal}$ to the BA between 1991 and 2024.

Water storage changes, in both liquid and solid form, impart both seasonal (periodic) and decadal changes to $g$. Surface processes, such as snow accumulation and melt, cause periodic changes and are much more straightforward to measure than subsurface ones, although gravimetry has been demonstrated to be useful in its monitoring on short and long timescales (Breili and Pettersen, 2009; Voigt et al., 2021). Annual freeze-melt cycles in the AL are also primarily periodic in their effects (Sect. 6.1.1).
Decadal progressive deepening (or thinning) of the AL could potentially be measurable with TLG. In periglacial environments in general, permafrost degradation will have measurable effects on $g$, as will groundwater storage changes (see Sect. 6.4). In RGs, specifically, intra-permafrost melt of excess ice results in a net loss of mass and may result in an increase in bulk density. At a given point in space above the RG, these processes would decrease $g$, although the effect on BA would be more complex to discern due to the potential for compaction and also due to dynamic effects from compressive thickening or extensive
thinning. Overall, our BA results suggest that, compared to 1991, there is now a relatively lower mass density in the central and frontal regions compared to the rooting zone. It is highly likely that both ice/water storage (Fig. 6) and rock movement processes have influenced these changes in the BA. Additionally, the periodic drying of frontal springs during melt periods (Sect. 6.1.2) supports the hypothesis that deep groundwater export may have also occurred. Future high-precision gravimetric measurements that can be compared in absolute terms to our 2024 surveys will enable the evaluation of long-term water/ice
storage change processes within the RG.

## 6.3   Potential of TLG for monitoring of permafrost and periglacial landforms

TLG offers clear opportunities for monitoring the various seasonal and long-term water/ice storage processes that occur in permafrost and periglacial environments. While gravimetry has existed for decades, modern, portable devices, such as the Scintrex CG-6 used in our study, offer levels of stability and accuracy that provide new opportunities for monitoring subsurface
processes whose effects may only be on the few $\mu$Gal level. Gravity measurements are non-invasive, integrative, and distributed,





*Landform and mass movements*

   Mass movements: Rock fall, debris flows, debris-laden avalanches

   Landform-scale permafrost creep

   Tectonic effects (Alpine uplift)

*Hydrological storage changes*

   Accumulation/melting of seasonal snowpack

   **Total subsurface water storage change** composed of:

   •Supra-permafrost (in AL) water storage changes

   •Supra-permafrost ice storage changes by seasonal ice build-up and melt
(periodic) and by permafrost degradation (progressive AL thickening)

   •Intra-permafrost ice/water storage changes (permafrost degradation, talik
formation)

   •Sub-permafrost groundwater storage changes in deeper aquifers (cirque
overdeepening) or fractured bedrock

**Table 3.** Types of mass changes or movement in mountain permafrost environment potentially affecting a time-lapse gravity signal. Periodic processes (e.g., AL freeze/thaw) will dominate on short (seasonal) time-scales, while other processes may have significant effects over longer time-scales.

and are possible essentially anywhere that is accessible by foot. Application of TLG to determine absolute gravity changes requires a known reference point, which may not always be feasible. Switzerland has a dense network of maintained absolute gravimetric reference points (Swisstopo, 2024) and the Murtèl RG has the particularity of being rapidly accessible via cable car for much of the year. This advantageous situation is not always the case for alpine sites. Nevertheless, there are also mult-

step strategies that can be employed to establish temporary local references, although they may require large travel distances and times, depending on the location. Finally, even relative TLG, wherein a fixed offset between surveys is not known, but the changes in $g$ from one point relative to another are, has already provided useful knowledge in the growing field of alpine hydrogeology (Arnoux et al., 2020).

The non-discriminatory sensitivity of gravimetry to mass changes can be an advantage in permafrost or periglacial environ-

ments, where water co-exists in solid and liquid state, as it yields measurements of the total water storage change including ice and water. Without the need for petrophysical relations, forward modelling can be relatively straightforward. This makes gravimetry an ideal complement to other geophysical measurements that are selective to specific physical properties (e.g., electro-magnetic or elastic properties). Conversely, attributing the total water storage change to individual supra-, intra-, and sub-permafrost ice/water storage changes (Table 3) is ambiguous and requires independent observations or *a priori* knowl-

edge of the stratigraphy and hydraulic properties. Gravimetric surveys also require precise positional measurements of survey points to enable accurate temporal comparisons. For example, in the case of subsidence caused by excess ice melt, mass is lost beneath the survey point, which should decrease gravity. Simultaneously, the subsiding point moves closer to Earth's centre





of mass, increasing gravity. Over multi-year timescales, the two competing effects can only be accounted for, and the loss of the excess ice can only be quantified, if altitude and VGG are known to high precision. As many complex processes occur in periglacial landforms over seasonal, yearly, and longer-term timescales, TLG surveys at multiple time-scales could provide valuable information to help constrain hydrological, structural, and rheological models of these geomorphological structures.

### 6.4 Permafrost–groundwater connectivity

As recently underlined in multiple recent reviews (Somers and McKenzie, 2020; Hayashi, 2020; van Tiel et al., 2024), alpine hydrogeological processes and, in particular, cryosphere–groundwater connectivity are insufficiently understood. In many alpine headwater catchments, RGs are a vital component of annual hydrological and hydrochemical dynamics (Brighenti et al., 2019). Permafrost ice, whether in RGs or other landforms such as frozen talus slopes (Mathys et al., 2024), is, in essence, immobile groundwater whose presence constitutes a water store and also modifies the subsurface hydraulic properties. Ice build-up and melt are, in turn, controlled as much by the ground thermal regime and the heat fluxes (Amschwand et al., 2024a) as by water availability from precipitation or snowmelt, be it the seasonal turnover in the AL or permafrost thaw in the underlying rock glacier core. Semi-impervious permafrost bodies can act as aquitards, leading to rapid supra-permafrost run-off of intense precipitation and a 'flashy' hydrograph typical of many permafrost-underlain catchments (Rogger et al., 2017). At Murtèl, however, the afore discussed hydrological measurements, stake measurements, and AL energy budgets at Murtèl RG suggest that up to $20-30\%$ of the snowpack is retained as AL ice and melts slowly enough to percolate through the semi-impermeable permafrost body. This 'leaky' behaviour and the hypothesis that coarse blocky permafrost landforms contribute to groundwater recharge by seasonally storing and routing winter precipitation to a sub-permafrost aquifer have been investigated at multiple sites (Amschwand et al., 2024b; Corte, 1987; Halla et al., 2021; Marchenko et al., 2024).

Due to its extensive documentation and overdeepened bowl-shaped bedrock base, Murtèl would be an ideal site for further studies investigating permafrost–groundwater connectivity. The concave underlying bedrock profile may facilitate the infiltration of basal flow to the fractured bedrock (Fig. 6). Characterisation of the fracture network in the granodiorite bedrock would allow for a more comprehensive understanding of the cryosphere–groundwater processes occurring at Murtèl and their implications for the wider catchment. A multi-method approach, combining hydrological and geophysical measurements with numerical hydrological modelling would help us to understand the timings, volumes and flow-paths of water originating from AL melt and permafrost degradation.

### 7 Conclusion and outlook

Climate change is inducing profound changes in the mountain cryosphere and is rapidly modifying the hydrology of high-mountain watersheds. Water percolation across permafrost bodies, as well as intermittent storage in the form of ice, controls groundwater recharge in permafrost-underlain watersheds. These cryo-hydrogeological processes contribute to headwater baseflow which is critical for aquatic habitats and water supply during droughts (Hayashi, 2020). Despite the importance of these phenomena, the complex links between the cryosphere and groundwater are poorly understood (van Tiel et al., 2024). There





is thus a strong need for quantitative and spatially-distributed methods for monitoring intra-annual, inter-annual and long-term changes in subsurface ice content and water storage in these environments. Recent technical advances in hydrogeodesy (Jaramillo et al., 2024) have opened new avenues for investigating the connectivity of groundwater with the rapidly-changing cryosphere. These advances complement established cryo-geophysical techniques (Hauck and Kneisel, 2008). Our application of time-lapse gravimetry (TLG) to estimate seasonal ice storage changes in the active layer of the Murtèl RG is the first use of this geophysical technique in a periglacial environment and has demonstrated the ability of the technique to spatially resolve AL thaw. Furthermore, despite the limitations of the historical data, our comparison with Vonder Mühll and Klingelé (1994) has shown the potential of multi-decade TLG measurements for permafrost degradation studies. With respect to our combined TLG–photogrammetry methodology and its application to an active RG, we observe that:

1. TLG is an ideal method for spatially-distributed quantification of mass movements and water/ice storage changes on seasonal scale due to its non-invasiveness, portability, and spatially-integrative sensitivity to mass changes.

2. The potential for long-term permafrost monitoring using TLG is promising, although comparison with historic gravimetry data is challenging unless the methodology is documented in detail and $\Delta g$ changes exceed the potentially large uncertainties of historic data. The initiation of high-precision gravimetry-photogrammetry (or gravimetry-LiDAR) surveys at new sites would provide an invaluable baseline for the monitoring of permafrost degradation and its associated hydrological changes.

3. In an environment as dynamic as the mountain cryosphere, digital surface models (DSMs) from rapid photogrammetric surveys are valuable–in many cases, necessary–for corrections of changing surface snow/névé/ice coverage and rock fall deposits between surveys. They could also serve to correct for landform movements in long-term TLG studies or those rapidly deforming ice-rich permafrost landforms.

Through continued interdisciplinary collaboration across the geophysics, geodesy, geomorphology, cryosphere, and hydrogeology research communities, we anticipate that methodological innovations and diverse perspectives will result in a deeper, more quantitative understanding of total water storage change processes in alpine environments. In mountain permafrost and periglacial environments, TLG holds significant potential for impact on its own and would provide unique, complementary data in joint deployments alongside other transient geophysical, geochemical, and hydrological monitoring techniques.

*Author contributions.* Both authors participated in all aspects of the study and wrote the manuscript collaboratively. LJSH obtained the funding and led the study. AI text generation tools were *not* used for any part of this manuscript.

*Competing interests.* The authors declare no competing interests.



*Acknowledgements.* This work is supported by the Swiss National Science Foundation via grant #212622 for the project "*RADMOGG*: Resilience and Dynamics of Mountain Groundwater using Gravimetry" (https://data.snf.ch/grants/grant/212622). We thank: Dani Vonder
525 Mühll (PHRT/ETHZ) for his encouragement and valiant attempts to retrieve the raw data from the 1991 surveys; Urs Marti (Bundesamt für Landestopografie - Swisstopo) for his generous assistance with the Bouguer anomaly calculations; Nazanin Mohammadi (University of Neuchâtel) for the *Gravi4GW_hybrid* and Earth tide correction script (which utilises *pyGtide* by Rau (2018)); University of Fribourg, WSL SLF and PERMOS for the webcam images; Corvatsch Bergbahnen for cable car transport; and Martin Andersen (UNSW) and Natalie Andersen for field assistance in the 09.2024 survey.



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
