# Peer review of "Investigating seasonal and multi-decadal water/ice storage changes in the Murtèl rock glacier using time-lapse gravimetry"

_EGUsphere, 2024_

## Author Response (AR1)

Dear Editor Mohammad Farzamian,

Thank you very much for your efficient handling of our manuscript. This was a thorough, yet positive, set of reviews with plenty of suggestions for small modifications, which we have implemented. Please find our documented edits below (in red).

Best regards,
-Landon Halloran and Dominik Amschwand

**Reviewer #1 (doi.org/10.5194/egusphere-2024-3933-RC1)**
This article describes a novel use of time lapse gravity to assess water storage changes from the active layer of a rock glacier in Switzerland. The paper is well written, has a comprehensive description of methodologies, and cites appropriate literature.
**We thank the reviewer for his/her time. We appreciate the positive overall assessment of our manuscript and the helpful comments. Likewise, we will make the edits as suggested, which will improve the clarity for readers of the final version.**

General comments for authors:
Only 8 gravity stations were reoccupied on and around the rock glacier for this study. Having more measurements would have provided more statistical significance to the interpretations of storage changes in different areas. Can the authors provide an explanation as to why only 8 stations? Can they recommend including more stations in future studies?
**In gravimetric surveys with a relative gravimeter, even a relatively stable one such as the CG6, it is necessary to return to reference points to correct for instrument drift. As an absolute reference, we chose a first-order station (highest precision type of reference points in the Swiss Gravity Network other than absolute stations (https://www.swisstopo.admin.ch/en/national-gravity-network). Travel between this reference station and the survey area involved driving, using a cable car with a fixed schedule, and hiking on the rock glacier. Due to time constraints, multi-day surveys were not possible. Thus, the number and location of the survey points were chosen to balance spatial coverage and a reasonably short survey duration. We will add a brief explanation of this in S3.2.**
**We have added this text: "The number of points and their locations were chosen so as to maximise coverage across the RG and to limit the survey duration, ensuring a better drift correction." … "Future surveys at Murtèl, or indeed at other RGs, could include more measurement locations, requiring multiple survey loops and a longer survey duration."**

Are there historical repeat lidar surveys that could help determine rates of subsidence or uplift of the rock glacier over time?
**Horizontal kinematic data are available from PERMOS. Estimates of vertical changes from laser scans and geodetic surveys suggest, however, small (mm/yr) or insignificant vertical changes.**

Figure 2 and Figure 3 could be improved by annotating the station numbers

*We will annotate as suggested.*

*Station numbers have been added to both.*

Detailed comments:

Page 2, line 46. I don't understand why there are 2 numbers for ice storage changes. Are these minimum and maximum changes over a period of years? The authors just need to clarify this a bit better.

*We will clarify.*

*We have added: …"for 2016-2017 and 2017-2018, respectively."*

Page 2, Line 54. Seismic is a broad category. Do the authors mean seismic refraction, seismic reflection, passive seismic monitoring? Should clarify.

*We will clarify.*

*To be more specific we have added the word "active" although this is just a list of some examples of methods (of which there are many) that have been used.*

Page 3, line 88. Is "slightly continental" the correct technical term? Consider rephrasing for accuracy

*We will rephrase.*

*"Moderately continental" now used*

Page 3, Line 91. I had trouble understanding what "permafrost-underlain" meant. Consider rephrasing to "Surficial material within the small (30 ha), non-glaciated Murtel catchment consists of unvegetated debris (on rock glaciers and talus slopes) and bedrock, with occurrences of permafrost".

*We will change the text for clarity.*

*Sentence now reads: "The small (30 ha), non-glacierised Murtèl catchment consists of rock glacier and talus slope debris and bedrock."*

Page 4, Line 94. What was measured in the bedrock depression? Consider rephrasing to "…depression and the buried bedrock topography has been mapped with a multi-geophysical…"

*We will change the text for clarity. The sentence now reads as suggested.*

Page 4, line 97. For clarity, consider changing "…that its thickness…" to "…that the AL thickness…"

*We will change the text for clarity. The sentence now reads as suggested.*

Page 4, Line 105. For clarity, consider changing "… distance, share this three-part stratigraphy" to "…distance, intercept this three-layer stratigraphy"

*We will change the text for clarity. The sentence now reads as suggested.*

Page 6, Line 135. How were the gravity observation points marked? With nails into rock? Spray paint? Please include this detail to help others trying this.

*We will add this info (paint).*

*The sentence now reads: "We marked survey points with paint and used a Leica GNSS DGPS system to survey all points to <2 cm absolute accuracy."*

Figure 2. Label the black dots with the station numbers.

**We will add this as per your earlier comment.**
*Station numbers have been added to both Figs 2 and 3.*

Page 10, line 237. Could the authors not calculate the snow volume change by subtracting the July DSM from the September DSM?
**This is essentially what we did (L234 in the pre-print), but it is slightly more complex as, in order to avoid noisy influences in non-snow covered areas, our process also involved smoothing and masking (as documented in S4.4).**
*This is already very thoroughly documented in S4.4.*

Page 10, line 262. Change "changes in the" to "changes were in the"
**We will modify it.**
*Edit applied: …"all net gravity changes were in the approximate range of"...*

Page 10, line 263. Add the detail that the -15.2 uGal change was for MUERTEL05
**We will add this.**
*Edit applied: …"from -15.2 uGal near the lower rooting zone (MURTEL05) to"…*

Page 10, line 264. Would "terminus" be a better word than "front"?
**We prefer the term "front", which is the more widely used term for rock glaciers.**

Figure 3. Add directions "S" and "N" to the plot in (a) to make it easier for the reader and label the 2024 points with the station numbers in both (a) and (b).
**We will modify the figure as suggested.**
*We have added station numbers to Fig 3a and 3b. As the survey points are not oriented exactly S-N, we refrain from adding this to Fig 3a. The orientation is clear when the reader considers both Figs 3a and 3b (which has an arrow and N direction marker) together.*

Page 14, Line 295. Change "comparisons on a relative" to "comparisons to the 2024 data on a relative"
**We will do this.**
*Edit made: …"which we use for comparisons to the 2024 data on a relative basis"…*

Page 14, line 296. Change "with an offset" to "with a DC offset"
**DC means direct current. We will use the term linear.**
*We decided "fixed offset" is more appropriate and applied this edit.*

Page 14, line 302. I did not see the ~200 uGal increase as described. Can this be annotated on Figure 3? Does it increase between the July and September surveys or from the smoothed trend of the 1991 points? Some clarification is needed here.
**We will clarify what we mean here (2024 vs 1991 trend line).**
*Edit applied: …"between the 1991 and 2024 surveys"...*

Figure 5. What are the dashed blue lines? Can you add this extra detail in the figure caption?

**We will add this.**

*We have rewritten this caption to improve its clarity:*
*"Figure 5. Calculated seasonal (08.07.2024–11.09.2024) below-ground total water storage change Δhwe for a range of hypothetical heff values (see Eqn. 5) and measured Δg values (thick lines) ±5 μGal (dotted lines) for the on-RG survey points MURTEL02-07 (see Fig. 4 for locations of survey points). The off-RG survey points MURTEL01 & 08 are included for completeness."*

Page 18, line 377. Change "aproportion" to "proportion"
**We will do this.**
*Edit made.*

Page 19, line 389. Instead of neighbouring 'regions', can you be more specific and say neighbouring 'catchments'?
**Regions is correct, some neighbouring regions are different catchments.**

Caption to Table 3. Change "environment potentially affecting" to " environments that potentially affect"
**We will do this.**
*We have improved this caption: …"or movement in mountain permafrost environments that potentially affect the surface value of $g$, and thus also time-lapse gravity signals."…*

Page 22, Line 464. Do the authors mean elevation instead of 'altitude'? Altitude is height above the local ground surface
**You're right. We will modify it.**
*Fixed: …"if VGG and elevation are"…*

**Reviewer #2 (doi.org/10.5194/egusphere-2024-3933-RC2):**
The paper concerns the novel use of time-lapse microgravimetry to investigate changes in water/ice storage in an already well studied rock glacier. The paper reads well, the topic is well introduced, and somehow becomes a review paper, thanks to the relevant and correct reference list. Some parts are perhaps too extended, as in Chapter 6.1, where the discussion is processed. I think that one of the main stability constraints remains in the gravimetric reference point near the site. This strengthens the results of the authors, but on the other hand may strongly limit the application of such a methodology elsewhere. This aspect is mentioned, but not emphasised as it should be, especially in chap. 6.3.

**We thank the reviewer for the time and effort he or she has put into providing us with an encouraging and thorough review. We believe that a good Introduction section should serve as a mini-review, so we are particularly happy to receive the comment on our completeness in introducing the topics.**

**Murtèl is, we agree, somewhat exceptional in that a high-quality absolute gravimetric reference station is accessible within a reasonably short travel-time. Regarding this limitation in potential applications to other rock glaciers, we have discussed this in the first paragraph of S6.3, but we will consider strengthening the language in the final manuscript.**

**While we believe the discussion of potential limitations of the application of TLG to RGs is already quite detailed (see the 2nd half of the first paragraph in S6.3), we have nonetheless expanded this discussion:**
**…"Application of TLG to determine absolute gravity changes requires a known reference point, which may not always be feasible. Switzerland has a dense network of maintained absolute gravimetric reference points (Swisstopo, 2024) and the Murtèl RG has the particularity of being rapidly accessible via cable car for much of the year. This advantageous situation is not always the case for alpine sites. Nevertheless, there are also multi-step strategies that can be employed to establish temporary local references, although they may require large travel distances and times, depending on the location. These constraints can effectively prevent the carrying out of absolute TLG surveys. In these cases, relative TLG, which involves using a reference point at which the value of g is not known absolutely and may vary with time, may be an option. Relative TLG, wherein a fixed Δg offset between surveys is not known, still provides data on changes in g at survey points relative to one another, has already provided useful knowledge in the growing field of alpine hydrogeology (Arnoux et al., 2020)."**

My main concern relates to the comparison with data from the 1990s. Although the authors correctly point out the great uncertainty of such an operation, I believe that this approach remains a great danger. Even though preliminary corrections are well explained, a simple but crucial argument like atmospheric pressure conditions is not considered. The authors rely on the atmospheric isolation of the CG-6 for their time series, but the same cannot be said for the data of the 90s. This aspect can strongly influence the resulting comparison and should at least be mentioned in the text. My suggestion is to limit the relevant results to the 2024 double surveys,

avoiding an ambitious multi-decadal approach. This would also significantly shorten the paper and reinforce the message of model TLG potentials by focusing on the authors' relevant results.

**Atmospheric pressure variations influence terrestrial gravity with an admittance of ~0.3 μGal/hPa (e.g., doi.org/10.1016/j.jog.2009.09.010). Under calm conditions, i.e., when gravimetric surveys are usually performed (windy days are avoided as they result in noisier data due to vibrations), daily variations in atmospheric pressure on the order of a few hPa are typical. Thus, the impact of atmospheric pressure would be on the order of 1 μGal. As this is well under the accuracy of any gravimeter available in the early 1990s, it is clear that atmospheric pressure effects can be safely ignored. Furthermore, gravity surveys, including those of Vonder Mühll & Klingelé (1994), are conducted in "loops", returning to a reference location in order to correct for drift. Thus, the corrected drift will include the effect of atmospheric pressure changes and consequently only the non-linear portion remains. This further reduces any residual effect of atmospheric pressure on the data. Consequently, while we thank the reviewer for raising these concerns, we unreservedly reject the idea of completely getting rid of the portion of our manuscript that compares our data with those of the 1991 survey. We will nonetheless ensure that we reinforce the cautious wording in our manuscript related to quantitative interpretation of the 2024 vs. 1991 results (this also links to your comment below re: Sections 3.4, 3.5, 5.2 & 6.2).**

*We have added text to S3.3: "Any residual effects of transient atmospheric pressure (Klügel and Wziontek, 2009) are also indirectly corrected for during the drift calculation."*

*We have also modified S5.2 and added text: "Comparisons of data from our surveys and the Vonder Mühll and Klingelé (1994) survey must be interpreted with caution due to the unquantified uncertainties in the 1991 data and the differences in the locations of our survey points. As such, we stress that the 1991--2024 comparison provides qualitative, relative insights rather than precise, quantitative measurement of Δg."*

Minor comments below:

Chap.2 the description of the site stratigraphy 91-95 is not clear and should be rewritten.

**We will modify this section for clarity (another reviewer also suggested this).**

*We have changed to: "The RG covers an elevation range from 2620 (base of front) to 2720 m asl. (transition to talus) (Fig. 1b–d). The talus slopes (2720–2800 m asl.) connect the active RGs to the headwalls that rise up to 3165 m asl. The small (30 ha), non-glacierised Murtèl catchment consists of unvegetated rock glacier and talus slope debris and bedrock. Murtèl is located at the permafrost margin and only its forefield (2600–2620 m asl.) is permafrost-free (Schneider et al., 2012, 2013). The rock glacier sits in a bowl-shaped, glacially overdeepened bedrock depression that has been measured with a multi-geophysical approach combined with borehole logs (Vonder Mühll and Klingelé, 1994)."*

Chap 3.1 statement on ice melt/gravity reduction needs to be better introduced, as you correctly state that several processes can explain the same results. I would avoid presenting this aspect here.

**The final sentence in S3.1 links the basic theory to the relevant cryo/hydro application. We will consider rephrasing.**

*We have modified this section and added a sentence: "After corrections to the raw data (Sect. 3.3), the residual effect on g due to mass distribution changes is revealed. Local mass loss due to ice melt or decreased groundwater storage in the vicinity below a measurement location will result in a decrease in gravity between two repeat surveys. As different mass change processes can result in the same Δg values at a given location, interpretation must be informed by other information (Sect. 6.3)."*

Chap 3.2 ln 133 <5h20m ?

**Less than 5 hours and 20 minutes.**

*We now spell it out: "under 5 hours, 20 minutes"*

Ln 135 what it means manually measured ?

**We will clarify this.**

*We have made the edit: …"measured with a measuring tape to millimeter precision"...*

Chap. 3.3 Atmospheric pressure is not given, neither for the historical nor for the recent measurements. I suppose the authors rely on the fact that cg-6 should be compensated, but they discuss temperature and tilt drift in detail without ever mentioning barometric issues. This should be clarified.

**We will mention barometric considerations (see comment above).**

*We have added text to S3.3 (see above).*

Chap 3.4;3.5;5.2;6.2 As explained, I find this temporal comparison too ambitious for quantitative estimation due to the large unresolved uncertainties.

**While we have already taken great care to highlight the limits of the 1991 dataset and to only draw qualitative conclusions when comparing the 1991 and 2024 data, we will reinforce the text as necessary in the revision to ensure readers do not try and draw quantitative conclusions.**

*We have added text to S5.2 (see above).*

Chap 5 Before presenting the results, the authors should logically present how they installed the gravimeter, as this is a crucial part of gravity measurements. For example, was the gravimeter installed on blocks? Digging snow? How was it fixed? Some pictures might help the readers, especially the non-experts.

**It is true that most readers of The Cryosphere will not be experienced in performing gravimetric surveys. We will add additional explanations (potentially including a photo figure) of these practical aspects.**

*Information on the methodology would logically fit into our methodology section (S3.2 concerns the actual gravimetry survey methodology). Thus, we have added a bit more detail there:*

*"The number of points and their locations were chosen so as to maximise coverage across the RG and to limit the survey duration, ensuring a better*

*drift correction. The total duration of each survey was under 5 hours, 20 minutes. Future surveys at Murtèl-or, indeed, at other RGs-could include more measurement locations, requiring multiple survey loops and a longer survey duration. We marked survey points with paint and used a Leica GNSS DGPS system to survey all points to <2 cm absolute accuracy. All survey points were located directly on bedrock or boulders. One challenge with gravimetric surveys on RGs is finding surfaces that are sufficiently horizontal so that the gravimeter can be levelled. Additionally, the height of the gravimeter above the ground surface was measured with a measuring tape to millimeter precision at each location."*

LN270-280 Again, no corrections for atmospheric changes are given.
**See above comments.**

Ln 313 typo (-)
**This is an "em dash"—a versatile and completely valid punctuation mark that is nowadays underused!**

Ln 328 double brackets
**We will fix this.**
*Fixed.*

Chap 6.4 In this clear hydrological presentation, talking about permafrost aquitards, I would also add https://doi.org/10.5194/tc-17-1601-2023
**Thank you for this reference. We have actually both already seen this paper and appreciated the approach the team carried out. We agree it makes sense to cite it in this section.**
*We have added this reference: "Semi-impervious permafrost bodies can act as aquitards (Pavoni et al., 2023), leading to rapid"...*

**Reviewer #3: Prof. Masaki Hayashi (doi.org/10.5194/egusphere-2024-3933-RC3)**
GENERAL COMMENTS

The manuscript presents an application of microgravimetry to estimate seasonal and long-term changes of water mass within a rock glacier. Using a case-study example, the authors present creative and innovative methods of data acquisition and analysis, which have the potential to provide a new tool for alpine permafrost research in broader regions around the world. The study will make significant contribution to advancing our understanding of rock glaciers and other permafrost landforms. The manuscript is very well written and organized. The data analysis is rigorous and figures are of high quality. I have a few minor suggestions to improve the clarity (please see below for specific comments). One important issue that needs a bit more careful attention is the attribution of observed mass changes to solid and liquid water. I feel that the authors dismiss the contribution of liquid water too casually. I will elaborate more on this in my specific comments below.

**We thank Prof. Hayashi for his thoughtful review of our manuscript. As he is one of the most well-known researchers working on alpine hydrogeology, receiving his positive assessment of our work is highly encouraging. We will make sure to give more weight to the role of groundwater as a Δg source in Section 6.4 in the final manuscript (see comments below). The majority of the considerations in our manuscript were focused on Murtèl, but it is important that we remain general in scope in the relevant parts of the Discussion.**
*We have added this text to S6.4: "TLG, as discussed in Sect. 6.3, is sensitive to all mass changes, and therefore the method would need to be combined with other techniques in order to resolve and differentiate storage changes in permafrost and groundwater."*
*We have also made further additions to the text as per your detailed comments below.*

SPECIFIC COMMENTS

Line 97. Its thickness. Does this refer to the thickness of the rock glacier or the active layer? I had to stop and think for a moment. Please explicitly state it.
**Another reviewer mentioned this, too. We will clarify.**
*The sentence now reads: "Geophysical investigations revealed that the AL thickness varies…"*

Line 113. The reader expects the 'Method' section here. I later discovered that both 3 and 4 are describing methods. Please change the section title to '3 Methods: Gravimetry' and '4 Methods: Snow ….'.
**Correct, we did split the methods into 2 separate sections. We will modify as suggested.**
*We have modified the section headings to "3. Methods: Gravimetry" and "4. Methods: Snow spatial distribution and corrections using aerial imagery".*

Line 216-218. Usually, there is a significant correlation between depth and density of snowpack. Therefore, depth-dependent snow density function

is commonly used in hydrological studies. It becomes evident later that the effect of snow is secondary to gravity measurements, but at this point the reader does not know how (in)significant it is. Please add a sentence to justify why a single value of density is used in calculation.

**We will clarify this in the final version.**

*We have added a bit more context and detail to this short section (S4.3): "In order to calculate the effect of snow on measured gravity, snow density needed to be measured. We measured the density using a portable digital balance (precision 0.1 g) and a coring tube (inner diameter 5 cm). The volume and mass of eighteen samples from different locations at various depths up to ~ 1m were measured. The measured sample sizes ranged from 137–736 cm3 and from 76–410 g. Measured densities ranged from 504–587 kg m–3, a typical range for névé (Fierz, C. et al., 2009). The average snow density was determined to be 550 ± 27 kg m–3."*

Figure 2. Please annotate '01', '02, … in Figure 2a, so the reader does not have to go back and forth between Figure 1, Figure 2, and Table 1.
Figure 3. Please annotate '05', '06', … in Figure 3b.

**Another reviewer also suggested this for Figs 2 and 3. We will annotate.**

*We have annotated Figures 2 and 3.*

Line 314. The authors attribute the 'dominant' gravity-change signal to ice storage. This is not unreasonable, but the justification is not convincing. Changes in the water table of supra-permafrost groundwater has been observed at numerous locations around the world, both in alpine and subarctic environments. Also, a quick glance of Figure 3 gives the impression that 08 is not likely underlain by ground ice. Please add a few sentences here to present a convincing justification for dismissing liquid water storage. The authors can also refer the reader to Figure 5h.

**Gravimetry is not selective on the type of water/ice changes. Several pieces of independent evidence do suggest that supra-permafrost water export is negligible on Murtèl on a seasonal scale:**

- **Surface outflow is flashy and not perennial once snowmelt is completed. After precipitation events in late summer, the supra-permafrost aquifer drains within a week and the rock glacier springs dry out. The active layer is coarse-blocky, on average not thicker than 2-3 meters, and inclined (10-12°), resulting in a small water retention capacity.**
- **Electrical resistivities of the Murtèl active layer are high and injecting current has been notoriously difficult, also suggesting a low water content.**

**Additionally, the estimated ice storage changes match 2021-2023 observations (below-ground stake measurements) of ice loss made in the active layer (and no reason to assume it was much different in 2024). We cite all relevant literature (the body of which is quite significant as Murtèl is well studied), but we will be sure to discuss these points more carefully, also noting that Murtèl is on the coarser end of typical Alpine rock glaciers.**

*We have added this text to section 6.1.1: "However, we caution that, on their own, the observed seasonal decreases in g cannot be attributed exclusively to active layer thaw, nor groundwater storage decreases."*

*And: "Due to its sparse fine-material content, the Murtèl AL has a small retention capacity for liquid water and supra-permafrost groundwater export is negligible on a seasonal scale. This interpretation is supported by the flashy outflow during thaw season (assuming a linear-reservoir behaviour). After precipitation events in late summer, the supra-permafrost aquifer drains rapidly within ~ 5 days and the rock glacier springs dry out (Amschwand et al., 2024b). Furthermore, electrical resistivities of the Murtèl active layer are high and injecting current has been notoriously difficult, also suggesting a low water content (Noetzli and Pellet, 2023). Since Murtèl is a coarse-blocky rock glacier, the effect of changing supra-permafrost groundwater levels is potentially important on rock glaciers having more fine material in the AL."*

*See also our reply above.*

Line 374-376. Groundwater export … is expected to be rather limited. From an objective reader's viewpoint, this sentence contradicts with Figure 5h, which clearly shows the drainage of groundwater. It is true that unweathered granodiorite has low hydraulic conductivity, but overlying sediments may have high enough conductivity, or the top few meters of granodiorite may be weathered. Please present more careful discussion of solid vs. liquid storage of water in both supra- and sub-permafrost zones. **We agree that groundwater export from a sub-permafrost aquifer or aquifer located away from the permafrost zone cannot be excluded based on our data. Indeed, it is highly likely that some degree of large-scale seasonal groundwater storage decline occurred over the Summer 2024 period, as is commonplace in alpine systems. The porosity of the granodiorite bedrock, which does have some degree of fracturation, is expected to be very small, meaning that changes in hydraulic head would result in relatively small Δg signals. Survey points 01 and 08 (the latter located in the permafrost-free forefield which has no sedimentary cover) are located off-RG. As gravimetry is a spatially integrative method, it is sensitive to mass changes that may occur with some lateral displacement relative to the measurement location. We will nonetheless reinforce the discussion around groundwater storage change contributing to measured Δg and also reiterate that TLG is a non-selective method.**

*The second-to-last paragraph in Section S6.1.2 has been significant expanded: "Assuming low hydraulic conductivity and porosity in the crystalline bedrock (granodiorite), groundwater export from the base of the Murtèl RG, lying in an over-deepened bowl-shaped depression (Barsch and Hell, 1975; Vonder Mühll and Klingelé, 1994) could reasonably be expected to be rather limited on the seasonal time-scale. Nonetheless, weathered bedrock at shallow depths may have significantly higher conductivities and porosities, allowing for seasonal storage changes that may affect g to a measurable level. The decreases in g at the off-RG points suggest that groundwater storage has decreased. While both MURTEL01 and MURTEL08 are likely to be influenced by RG active layer thaw, MURTEL08 in particular (Fig 5h) suggests that either groundwater storage decreases have occurred. There is, however, no borehole intersecting the bedrock, thus it is impossible to draw firm conclusions about groundwater storage changes at Murtèl where known seasonal active layer thaw occurs and has been directly measured."…*